# Red Blood Cell DHA Is Inversely Associated with Risk of Incident Alzheimer’s Disease and All-Cause Dementia: Framingham Offspring Study

**DOI:** 10.3390/nu14122408

**Published:** 2022-06-09

**Authors:** Aleix Sala-Vila, Claudia L. Satizabal, Nathan Tintle, Debora Melo van Lent, Ramachandran S. Vasan, Alexa S. Beiser, Sudha Seshadri, William S. Harris

**Affiliations:** 1Fatty Acid Research Institute, Sioux Falls, SD 57106, USA; nlt@faresinst.com (N.T.); wsh@faresinst.com (W.S.H.); 2Cardiovascular Risk and Nutrition, IMIM (Hospital del Mar Medical Research Institute), 08003 Barcelona, Spain; 3Glenn Biggs Institute for Alzheimer’s and Neurodegenerative Diseases, UT Health San Antonio, San Antonio, TX 78299, USA; satizabal@uthscsa.edu (C.L.S.); melovanlent@uthscsa.edu (D.M.v.L.); seshadri@uthscsa.edu (S.S.); 4Department of Population Health Sciences, UT Health San Antonio, San Antonio, TX 78229, USA; 5The Framingham Heart Study, Framingham, MA 01702, USA; 6Department of Neurology, Boston University School of Medicine, Boston, MA 02118, USA; 7Department of Statistics, Dordt University, Sioux Center, IA 51250, USA; 8Schools of Medicine and Epidemiology, Boston University, Boston, MA 02118, USA; vasan@bu.edu; 9Department of Biostatistics, Boston University School of Public Health, Boston, MA 02118, USA; alexab@bu.edu; 10Sanford School of Medicine, University of South Dakota, Sioux Falls, SD 57069, USA

**Keywords:** omega-3, brain health, neurodegeneration, lipids, elders

## Abstract

Docosahexaenoic acid (DHA) might help prevent Alzheimer’s disease (AD). Red blood cell (RBC) status of DHA is an objective measure of long-term dietary DHA intake. In this prospective observational study conducted within the Framingham Offspring Cohort (1490 dementia-free participants aged ≥65 years old), we examined the association of RBC DHA with incident AD, testing for an interaction with *APOE-ε4* carriership. During the follow-up (median, 7.2 years), 131 cases of AD were documented. In fully adjusted models, risk for incident AD in the highest RBC DHA quintile (Q5) was 49% lower compared with the lowest quintile (Q1) (Hazard ratio [HR]: 0.51, 95% confidence interval [CI]: 0.27, 0.96). An increase in RBC DHA from Q1 to Q5 was predicted to provide an estimated 4.7 additional years of life free of AD. We observed an interaction DHA × *APOE-ε4* carriership for AD. Borderline statistical significance for a lower risk of AD was observed per standard deviation increase in RBC DHA (HR: 0.71, 95% CI: 0.51, 1.00, *p* = 0.053) in *APOE-ε4* carriers, but not in non-carriers (HR: 0.85, 95% CI: 0.65, 1.11, *p* = 0.240). These findings add to the increasing body of literature suggesting a robust association worth exploring dietary DHA as one strategy to prevent or delay AD.

## 1. Introduction

Evidence that dietary factors can influence risk for Alzheimer’s disease (AD) continues to accumulate [1]. Specifically, docosahexaenoic acid (22:6n−3, DHA), which is naturally found in fatty fish, is an omega-3 fatty acid selectively enriched in membrane phospholipids of the central nervous system [2]. Experimental studies reported that DHA ameliorates several AD-associated features, including amyloid-beta peptide aggregation into oligomers and fibrils [3], brain glucose hypometabolism [4], and neuroinflammation [5]. These observations support the view that an increased intake of DHA might lower risk for developing AD, particularly in at-risk populations such as carriers of the *APOLIPOPROTEIN E (APOE)-ε4* allele [6]. Given the marginal de novo synthesis of DHA [2], measurement of circulating or tissue levels of DHA is a valid biomarker of dietary DHA intake, which allows one to circumvent the uncertainties of self-reported dietary data [7]. Given the red blood cell (RBC) lifespan (around 120 days), RBC DHA better reflects long-term DHA intake than the DHA content of other blood lipid pools, including total serum/plasma or serum/plasma phospholipids [7], with a within-person stability documented over at least a 6-week period [8].

Despite the encouraging results from epidemiological studies on RBC DHA status and cognition (reviewed in [9]) and the findings of neuroimaging studies (reviewed in [9,10]), data on RBC DHA status and incident AD are scarce. No significant associations of RBC DHA and incident dementia and AD were observed in a prospective study conducted in 663 Canadians over 65 years old with a median follow-up of 4.9 years [11]. Interestingly, a higher blood mercury level—another biomarker of fish intake—was associated with a lower risk for dementia in this study. In a much larger cohort of 65-year-old women with a median follow-up of 9.8 years, an 8% decrease (per 1-SD higher RBC DHA, *p* = 0.08) in risk for probable dementia was observed [12]. Other prospective studies dealing with total serum/plasma or serum/plasma phospholipids either did [13,14,15,16] or did not [17,18] report that circulating DHA was related to a lower risk of AD/dementia.

The limited data on the topic, along with the emerging interest on the interaction between *APOE* genotype and DHA [6] warrants further analysis of the question using well-conducted prospective studies with an adequate statistical power. Here, we hypothesized that in participants aged 65 years and older, higher RBC levels of DHA are associated with lower risk of incident AD and of all-cause dementia, and that an interaction with *APOE-ε4* carriership exists. We addressed this question in a prospective setting from the Framingham Offspring Cohort, which was followed for neurological events for up to 14 years after RBC DHA was measured.

## 2. Materials and Methods

### 2.1. Study Population

The Framingham Health Study involves ongoing population-based cohorts from the town of Framingham, Massachusetts, USA. The Original cohort was established in 1948 with the aim to identify factors that contribute to the development of cardiovascular disease [19]. In 1971, the Offspring cohort was established, including children of the original cohort and their spouses [20]. The Offspring cohort enrolled 5124 participants who have been studied over nine examination cycles, approximately once every 4 years. Of the original Framingham Offspring Cohort, 3021 attended their eighth examination cycle (2005–2008), at which RBCs were collected and ultimately analyzed for DHA content. Participants were excluded in hierarchical order if they were missing RBC fatty acid measurements (*n* = 143), had prevalent dementia at exam 8 (*n* = 27), underwent the most recent dementia assessment before RBC obtention (*n* = 19), were <65 years old (*n* = 1297), or were missing information on *APOE* genotype (*n* = 45) [21]. These criteria eliminated 1531 participants, leaving 1490 eligible for the present investigation (Figure 1).

### 2.2. RBC Fatty Acid Analysis

Blood was drawn after a 10–12 h fast into an EDTA tube; RBCs were isolated by centrifugation and were frozen at −80 °C immediately after collection. RBC fatty acid composition was determined as described previously [22]. Briefly, RBCs were incubated at 100 °C with BF_3_-methanol and hexane to generate fatty acid methyl esters that were then analyzed by gas chromatography with flame ionization detection (Figure 2). Twenty-seven fatty acids were quantified and DHA was expressed as a percent of total RBC fatty acids. The inter-assay coefficient of variation for DHA was 4.9%.

### 2.3. Outcome Measures

Our primary outcome was incident AD developing at any time after the eighth examination cycle to the end of 2018. Our secondary outcome was all-cause dementia. The median follow-up time was 7.2 years (range 0–14 years). From examination cycle seven onward, all participants were invited to complete neurocognitive testing. Participants were flagged with suspected cognitive impairment using the Mini-Mental State Examination if (1) performance fell below education-based cutoff scores [23]; (2) a decline of 3 or more points was observed between consecutive examinations, or (3) a decrease of 5 or more points was observed from the participant’s highest past Mini-Mental State Examination score. Participants were also flagged following referrals or concern expressed by the participant, their family, or primary care physician. In this case, additional yearly neuropsychological assessments were performed between the quadrennial Offspring examinations. Adjudication of dementia and dementia subtypes, and date of diagnosis, was reached by a committee comprising of at least one neuropsychologist and one neurologist after a detailed review of all available neurological examination records, neuropsychological assessments, neuro-imaging data, hospital/nursing home/outpatient clinic records, information from family interviews, and autopsy results (when available). A diagnosis of dementia was based on the Diagnostic and Statistical Manual of Mental Disorders (4th edition) criteria requiring impairment in memory and at least one other domain of cognitive function, along with impaired functional ability. A diagnosis of AD dementia was reached based on the criteria of the National Institute of Neurological and Communicative Disorders and Stroke and the AD and Related Disorders Association for definite, probable, or possible AD [24]. For individuals without known incident events, follow-up was censored at the time of death or the date the participant was last known to be cognitively normal, through December 2018.

### 2.4. Statistical Analyses

Categorical variables were expressed as frequencies and percentages, whereas quantitative variables were expressed as means (95% CI). Baseline differences between *APOE-ε4* carriers and non-carriers were assessed by 1-factor ANOVA or the chi-square test, as appropriate. The associations between RBC proportion of DHA and risk of incident AD and all-cause dementia (two separate outcomes) were examined using multivariable Cox proportional hazards models. Schoenfeld residuals were used to confirm the proportional hazards assumption. Results are reported as hazard ratios (HR) with corresponding 95% confidence intervals (CI). HRs were estimated on a per quintile basis compared to the lowest quintile, as well as for a linear trend across quintiles. We also calculated HRs and 95% CIs associated with a 1-SD increment in RBC DHA (i.e., 1.4% of RBC fatty acids). We computed models using different sets of covariates. In Model one, we adjusted for non-modifiable risk factors, including age, sex, and *APOE-ε4* carriership. In Model two, we adjusted for the variables in Model one plus education and diabetes status at baseline. In Model three, we adjusted for all Model two variables plus prevalent cardiovascular disease at baseline. We estimated cumulative incidence and plotted DHA Q1 vs. Q5 and AD, as well as all-cause dementia. We also replaced RBC DHA with RBC eicosapentaenoic acid (EPA, 20:5n−3), and the omega-3 index (RBC DHA + EPA) as exposures of interest given that similar studies documented significant associations for these exposures [12,17,18]. Finally, we estimated effects on all-cause dementia- or AD-free years of life associated with a Q5 to Q1 change in RBC DHA and compared this with the estimated effects of increasing age at baseline and *APOE-ε4* carriership. In secondary exploratory analyses, we tested for interactions between RBC DHA and *APOE-ε4* carriership in association with incident AD and all-cause dementia. Because of the well-known opposing effects of *ε2* (protective) and *ε4* (harmful) alleles for AD, this analysis was conducted after exclusion of *ε2/ε4* participants (*n* = 44). We adjusted for age, sex, education, prevalent diabetes, and prevalent CVD at baseline. We used a significance level of α = 0.10, and in case of a statistically significant interaction, we then stratified the sample by *APOE-ε4* carriership to further search for group-specific associations. Splines relating RBC DHA and risk for incident disease were fit on the combined sample, as well as in *APOE-ε4* carriership strata with third-degree polynomials using the splines package in R (version 3.6.2). Adjusted models used a linear model to separately predict AD and all-cause dementia risk by third-degree polynomial splines with knots at tertiles of DHA. Unless otherwise stated, the significance level α = 0.05. We completed all analyses using R (version 3.6.2).

## 3. Results

Table 1 shows the characteristics of the study population.

The associations between RBC DHA and the risk of the two clinical endpoints are presented in Table 2.

In fully adjusted models, compared to participants at lowest quintile of RBC DHA (Q1 < 3.8%, median DHA = 3.4%), those at the top quintile (Q5 > 6.1%, median DHA = 7.0%) had a 49% reduction in risk for incident AD (HR: 0.51, 95% CI: 0.27, 0.96). For all-cause dementia, risk was reduced by 44% (HR: 0.56; 95% CI: 0.32, 0.97). When analyzed as a continuous variable (per SD increase in RBC DHA), a lower risk for AD was observed in Model 1 (*p* = 0.030), but this was modestly attenuated in Models 2 (*p* = 0.060) and 3 (*p* = 0.052). For all-cause dementia, change per SD was not significant, although linear trend tests that trended towards reduced risk were observed in all three models (Model 1, *p* = 0.062; Model 2, *p* = 0.093, and Model 3, *p* = 0.079). In a sensitivity analysis eliminating all diagnoses within 5 years of baseline, similar trends were observed (Table 3).

Figure 3 shows the crude cumulative AD and all-cause dementia incidence among participants in Q1 and Q5, while Figure 4 uses splines to illustrate the overall risk reduction in both AD and all-cause dementia.

Analysis using the RBC EPA and omega-3 index showed slightly attenuated risk estimates compared to RBC DHA alone (Table 4 and Table 5, respectively).

Theoretical effects on years of life free of AD and all-cause dementia per a Q5 to Q1 difference in RBC DHA were compared with the estimated effects of age (per 1 year older at baseline) and *APOE-ε4* (compared with non-carriers) (Table 6). A low (Q1) versus a high (Q5) RBC DHA was associated with an estimated reduction of 4.65 years free of AD and 4.03 years free of all-cause dementia, respectively. Carrying an *APOE-ε4* allele (versus not) was associated with a reduction of 7.59 and 7.30 years free of AD and all-cause dementia, respectively.

In a secondary analysis of the effects of *APOE-ε4* carriership on the RBC DHA relationship with all-cause dementia and AD, in three of the four interactions, our exploratory statistical significance threshold of 0.10 was met. Regarding AD, there was evidence of an interaction between DHA levels and *APOE-ε4* carriership for both the continuous model (*p* = 0.093) and non-linear splines (*p* = 0.100). Regarding all-cause dementia, there was evidence of an interaction between the non-linear spline (*p* = 0.091), but not for the continuous model (*p* = 0.280). In stratified analyses, a trend towards a lower risk of AD was observed per SD increase in RBC DHA for the linear trend in *APOE-ε4* carriers (HR: 0.71, 95% CI: 0.51, 1.00, *p* = 0.053), but not in non-carriers (HR: 0.85, 95% CI: 0.65, 1.11, *p* = 0.240). Splines by *APOE-ε4* carriership strata are depicted in Figure 5.

## 4. Discussion

In this prospective study conducted in a community-based sample of Americans over age 65 who were followed for a median of 7.2 years for incident dementia, we found that an increasing proportion of DHA in RBCs was related to a lower risk of AD and all-cause dementia. Of note, participants at the top quintile of RBC DHA had roughly half the risk of developing AD during follow-up compared to those at bottom quintile. We also detected a possible interaction between RBC DHA × *APOE-ε4* carriership, with a stronger inverse association between RBC DHA and risk of AD in *ε4* carriers—individuals at increased genetic risk of late-onset AD—than non-carriers. This suggests that carriers may benefit more from higher DHA levels than non-carriers [6].

Three of our findings are important. First, this study supports the hypothesis of a link between diet and brain health, since the most effective way to raise RBC DHA levels is to consume more preformed DHA. Thus, DHA, a fatty acid also known to have cardiovascular benefits [25], might also slow the progression of AD. Second, based on our estimations, changing from the lowest quintile (<3.8% of DHA in RBC membranes) to the top quintile (>6.1%) could translate into an estimated gain of 4.7 years free of AD. This was roughly half of the apparent benefit gained from not carrying an *APOE-ε4* allele. Given that estimated health-care payments in 2021 for all patients with AD or other dementias amount to $355 billion in US (not including caregiving by family members and other unpaid caregivers) [26], any cost-effective strategy for delaying the onset of AD is of utmost public health interest. Delaying AD by 5 years leads to 2.7 additional years of life, and 4.8 additional AD-free years for an individual who would have acquired AD, and is worth over $500,000 [27]. Third, after excluding *ε2/ε4* participants (because of the known protective effects of the *ε2* allele), we observed an interaction DHA × *APOE-ε4* carriership on incident AD and all-cause dementia, with a trend towards a greater benefit of DHA in *ε4* carriers than in non-carriers. A plausible explanation for this finding is that *APOE-ε4* carriers might need more DHA to overcome their lower status of DHA (secondary to accelerated liver catabolism of DHA) coupled to impaired delivery of DHA to the brain [6]. This exploratory finding, which should be confirmed in more prospective studies with adequate statistical power, suggests that the *APOE* genotype modulates the associations between DHA and incident AD, and reinforces the need to target these particular individuals for supplementation, as expanded upon below.

Our study is in line with that of Tan et al., who reported cross-sectional associations with RBC DHA on cognitive performance and brain volume measurements (with higher DHA being associated with beneficial outcomes) in the same cohort as studied here [28]. Most interestingly, 15 years ago, similar findings were reported by Schaefer et al. in the parents of the individuals who were the focus of this present investigation (i.e., the Original Framingham Heart Study cohort). Schaefer et al. reported that participants in the top quartile of plasma phosphatidylcholine DHA experienced a significant, 47% reduction in the risk of developing all-cause dementia compared with those with lower levels [13]. Similar findings a generation apart in a similar genetic pool provide considerable confirmation of this DHA–dementia relationship. 

Despite mounting evidence on the association between circulating DHA and preserved brain structure [9,10], blood–brain barrier integrity [29], and lower cerebral amyloidosis [30], several longitudinal studies on circulating DHA and incident AD/dementia failed to report statistically significant associations for DHA [11,12,17,18], while reporting significant inverse associations for DHA + EPA [12,15] or EPA alone [17,18]. In our study, using RBC EPA + DHA (i.e., the omega-3 index) or RBC EPA as the exposures of interest resulted in weaker and non-significant associations than for DHA alone. Future research is warranted to better delineate the extent to which EPA and/or DHA is the better marker of risk for dementia, and whether plasma concentrations vs. percent composition vs. RBC is the optimal sample type to analyze for omega-3 content when evaluating patients with respect to dementia.

In terms of clinical relevance, the lack of benefits in cognitive performance in randomized controlled trials involving DHA [31,32,33,34,35] urges to improve the design of future trials. Other study designs to elucidate causation (e.g., Mendelian Randomization) may also be valuable, though identifying a good quality genetic instrument for DHA may prove challenging [36,37]. Our results imply that certain people might benefit more from DHA-based interventions than others. This perspective is aligned with the 21st century shift towards “precision nutrition” and “personalized medicine.” Specifically, two patient characteristics would be of interest. First, those with low DHA status, as suggested by results from the Multidomain Alzheimer Preventive Trial (MAPT), in which 3-year supplementation with 800 mg DHA + 225 mg EPA showed no significant effect on cognitive decline overall in older people with memory complaints [34], but benefits were observed in a subgroup of individuals with low omega-3 status at baseline [38]. This finding spawned the ongoing “low-omega (LO)-MAPT” trial (18-month intervention in older adults with omega-3 index ≤ 4.83%; [39]), which will hopefully shed light on this issue. The second group that might benefit from DHA supplementation is individuals who are genetically at risk of AD (i.e., *APOE-ε4* carriers). In these people, subclinical structural and functional brain changes associated with AD take place years (even decades) before AD is present. There is increasing evidence for cognitive benefit from dietary DHA in cognitively healthy ε4 carriers (consistent with our findings), but not in those with AD or mild cognitive impairment [6]. Therefore, there may be a window of opportunity to identify cognitive healthy *ε4* carriers and manage their associated elevated dementia risk with a dietary intervention (i.e., dietary DHA, but requiring doses close to 2 g/d [40]).

The strengths of this study are the inclusion of a large sample of older adults living in a community setting, with comprehensive cognitive assessments, continuous dementia surveillance, and collection of multiple health measures that can be included as potential confounders in statistical models. Furthermore, we used objective measurements of DHA and EPA from RBC, which reflect their long-term intake more accurately than dietary intake questionnaires. However, our study has several limitations. First, given its observational nature, it cannot address causality, and it is not possible to establish the directionality of associations. Second, the low number of *ε4* carriers resulted in a less precise effect estimates; therefore, our exploratory finding should be replicated in larger studies with greater statistical power. Third, we could not exclude the possibility that uncaptured environmental or other genetic factors may have influenced or caused the observed associations. Fourth, there is no information on whether a single measurement of RBC DHA is appropriate to estimate the risk of AD over long-term follow up when compared to repeated measurements. Finally, additional studies are needed to replicate these results in more diverse populations.

## 5. Conclusions

In conclusion, in a cohort of dementia-free participants from the Framingham Heart Study aged 65 years and older, we observed that those with a baseline RBC DHA proportion above 6.1% (top quintile) had nearly half the risk of developing AD (and all-cause dementia), and had an estimated 4.7 extra years of life free of AD compared to those with an RBC DHA below 3.8% (bottom quintile). In addition, we observed a trend for a stronger association in between RBC DHA and risk for dementia in *ε4* carriers than non-carriers, a finding that needs further research. Our results, which concur with a growing experimental research foundation, suggest that an increased DHA intake may be a safe and cost-effective strategy in preventing AD in specific populations.

## Figures and Tables

**Figure 1 nutrients-14-02408-f001:**
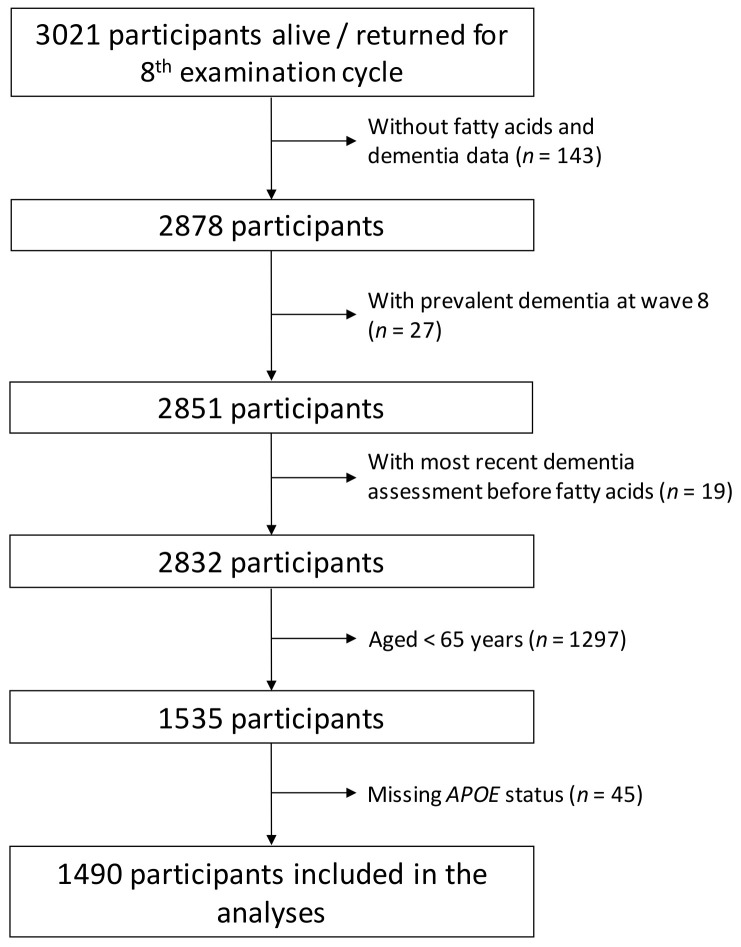
Flowchart of the study.

**Figure 2 nutrients-14-02408-f002:**
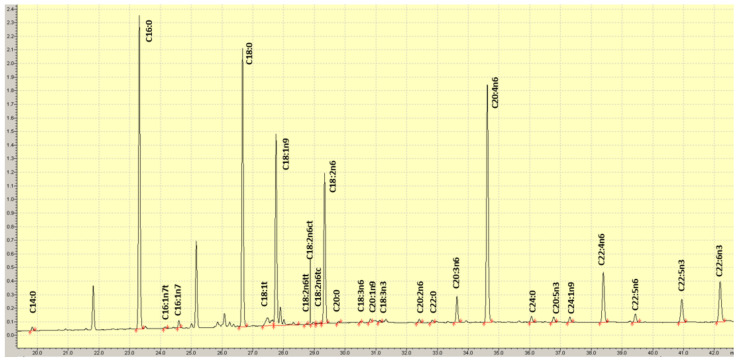
Example of a chromatogram of RBC fatty acid methyl esters.

**Figure 3 nutrients-14-02408-f003:**
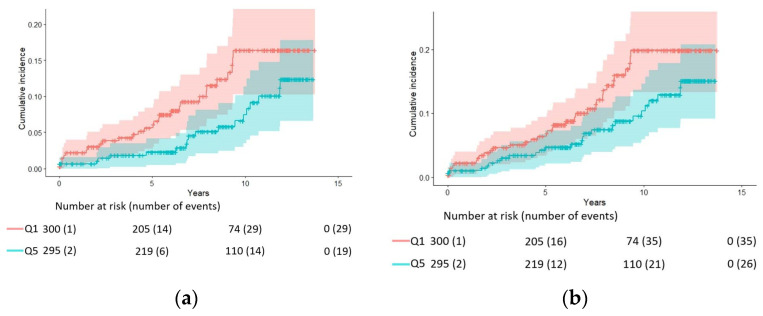
Crude cumulative Alzheimer’s disease (*p* = 0.04) (**a**) and all-cause dementia (*p* = 0.07) (**b**) in participants with baseline red blood cell DHA in the upper quintile (Q5) compared with those in the lowest one (Q1).

**Figure 4 nutrients-14-02408-f004:**
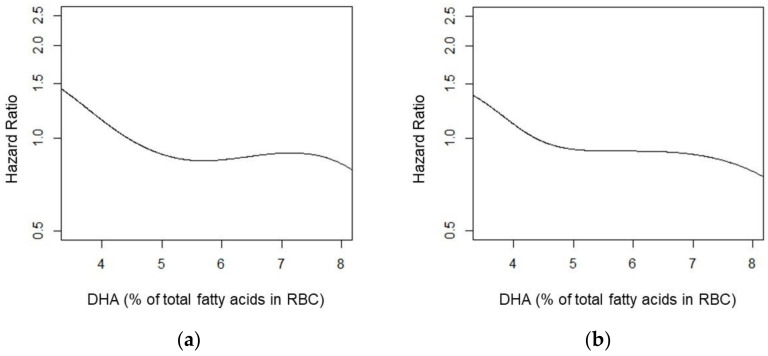
Spline for red blood cell DHA and Alzheimer’s disease (**a**) and all-cause dementia (**b**). Model adjusted for age, sex, *APOE-ε4* carriership, education, diabetes status at baseline, and prevalent cardiovascular disease at baseline.

**Figure 5 nutrients-14-02408-f005:**
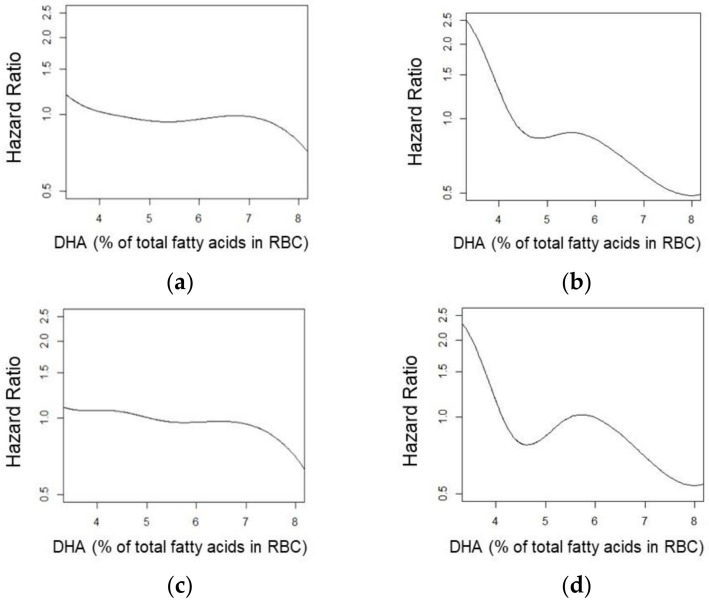
Splines by strata (*n* = 1466) for Alzheimer’s disease (**a**) and all-cause dementia (**b**) according to the absence (**c**) or presence (**d**) of *APOE-ε4*.

**Table 1 nutrients-14-02408-t001:** Baseline characteristics and endpoints of the study population, overall and by *APOE-ε4* carriership.

Variable	All Population (*n* = 1490)	*APOE-ε4* Non-carrier (*n* = 1155) ^1^	*APOE-ε4* Carrier (*n* = 335) ^2^	*ε3/ε4* + *ε4/ε4* (*n* = 311)
Women—No. (%)	798 (53.6)	620 (53.7)	178 (53.1)	167 (53.7)
Age—Mean (SD)	73.0 (5.7)	73.2 (5.8)	72.4 (5.5) ^3^	72.5 (5.5)
Education—No. (%)				
No high school degree	64 (4.3)	49 (4.2)	15 (4.5)	15 (4.5)
High school degree	341 (22.9)	275 (23.8)	66 (19.7)	62 (20.0)
Some years of college	366 (24.6)	282 (24.4)	84 (25.1)	77 (24.8)
College degree	505 (33.9)	384 (33.2)	121 (36.1)	112 (36.0)
Unknown	214 (14.4)	165 (14.6)	49 (14.6)	45 (14.5)
Diabetes—No. (%)	256 (17.2)	202 (17.5)	54 (16.1)	53 (17.0)
Prevalent cardiovascular disease—No. (%)	330 (22.1)	252 (21.8)	78 (23.3)	74 (23.8)
RBC DHA, proportion of total fatty acids—Mean (SD)	5.00 (1.37)	5.00 (1.37)	5.00 (1.39)	5.04 (1.40)
Incident Alzheimer’s disease—No. (%)	131 (8.8)	77 (6.7)	54 (16.1) ^3^	51 (16.4) ^3^
Incident all-cause dementia—No. (%)	168 (11.3)	103 (8.9)	65 (19.4) ^3^	61 (19.6) ^3^

^1^ Includes *ε2/ε2* (*n* = 7), *ε2/ε3* (*n* = 191), and *ε3/ε3* (*n* = 957). ^2^ Includes *ε2/ε4* (*n* = 24), *ε3/ε4* (*n* = 286), and *ε4/ε4* (*n* = 25). ^3^
*p* < 0.05 vs. non-carrier. DHA, docosahexaenoic acid; RBC, red blood cell.

**Table 2 nutrients-14-02408-t002:** Hazard ratios (HR) for red blood cell DHA on Alzheimer’s disease (AD) and all-cause dementia (*n* = 1490).

Endpoint	HR (95% CI) for Quintiles of Red Blood Cell DHA						HR (95% CI) per SD ^1^
	Q1 (<3.8%, Median = 3.4%) (*n* = 300)	Q2 (3.8% to <4.5%, Median = 4.2%) (*n* = 298)	Q3 (4.5% to <5.2%, Median = 4.8%) (*n* = 297)	Q4 (5.2% to 6.1%, Median = 5.6%) (*n* = 297)	Q5 (>6.1%, Median = 6.97%) (*n* = 295)	*p* for Trend	
AD							
N. of cases	29	30	24	29	19		131
Model 1	1.00	0.72 (0.43, 1.21)	0.61 (0.34, 1.09)	0.72 (0.41, 1.25)	0.48 (0.26, 0.87)	0.04	0.80 (0.66, 0.98)
Model 2	1.00	0.78 (0.45, 1.33)	0.64 (0.35, 1.18)	0.75 (0.42, 1.35)	0.51 (0.27, 0.98)	0.07	0.82 (0.67, 1.01)
Model 3	1.00	0.77 (0.45, 1.33)	0.64 (0.35, 1.18)	0.75 (0.42, 1.33)	0.51 (0.27, 0.96)	0.06	0.82 (0.66, 1.00)
Dementia							
N. of cases	35	38	29	40	26		168
Model 1	1.00	0.78 (0.49, 1.26)	0.64 (0.38, 1.08)	0.87 (0.54, 1.41)	0.56 (0.33, 0.94)	0.09	0.85 (0.72, 1.01)
Model 2	1.00	0.80 (0.49, 1.29)	0.65 (0.38, 1.11)	0.88 (0.54, 1.45)	0.57 (0.33, 0.99)	0.12	0.86 (0.73, 1.03)
Model 3	1.00	0.79 (0.49, 1.29)	0.64 (0.37, 1.11)	0.87 (0.53, 1.44)	0.56 (0.32, 0.97)	0.10	0.86 (0.72, 1.02)

^1^ 1 SD = 1.4% of total fatty acids. CI = confidence interval; Q = quintile; SD = standard deviation. Model 1, adjusted for age, sex, and *APOE-ε4* carriership (non-carrier vs. carrier); model 2, further adjusted for education (no high-school degree vs. high school degree vs. some years of college vs. college degree vs. unknown) and baseline diabetes (yes vs. no); model 3, further adjusted for prevalent cardiovascular disease at baseline (yes vs. no).

**Table 3 nutrients-14-02408-t003:** Hazard ratios (HR) for red blood cell DHA on Alzheimer’s disease (AD) and all-cause dementia (*n* = 1453) ignoring 37 individuals with incident Alzheimer’s or all-cause dementia within five years of baseline.

Endpoint	HR (95% CI) for Quintiles of Red Blood Cell DHA						HR (95% CI) per SD ^1^
	Q1 (<3.8%, Median = 3.4%) (*n* = 284)	Q2 (3.8% to <4.5%, Median = 4.2%) (*n* = 282)	Q3 (4.5% to <5.2%, Median = 4.8%) (*n* = 284)	Q4 (5.2% to 6.1%, Median = 5.6%) (*n* = 280)	Q5 (>6.1%, Median = 6.97%) (*n* = 283)	*p* for Trend	
AD							
N. of cases	15	18	9	17	13		72
Model 1	1.00	0.82 (0.41, 1.66)	0.41 (0.17, 0.99)	0.81 (0.38, 1.73)	0.61 (0.28, 1.32)	0.29	0.84 (0.64, 1.11)
Model 2	1.00	0.82 (0.40, 1.69)	0.40 (0.16, 0.95)	0.82 (0.37, 1.82)	0.57 (0.24. 1.34)	0.29	0.84 (0.62, 1.12)
Model 3	1.00	0.80 (0.39, 1.65)	0.38 (0.16, 0.92)	0.77 (0.35, 1.73)	0.53 (0.22, 1.26)	0.23	0.81 (0.61, 1.09)
Dementia							
N. of cases	19	22	13	23	14		91
Model 1	1.00	0.81 (0.43, 1.53)	0.49 (0.24, 1.03)	0.91 (0.48, 1.76)	0.53 (0.26, 1.09)	0.18	0.83 (0.66, 1.05)
Model 2	1.00	0.77 (0.41, 1.46)	0.47 (0.23, 0.98)	0.89 (0.45, 1.76)	0.49 (0.23, 1.04)	0.17	0.82 (0.64, 1.05)
Model 3	1.00	0.75 (0.39, 1.42)	0.45 (0.21, 0.95)	0.84 (0.43, 1.67)	0.45 (0.21, 0.97)	0.11	0.81 (0.63, 1.02)

^1^ 1 SD = 1.4% of total fatty acids. CI = confidence interval; Q = quintile; SD = standard deviation. See Table 2 for detailed information on statistical models.

**Table 4 nutrients-14-02408-t004:** Hazard ratios (HR) for red blood cell EPA on Alzheimer’s disease (AD) and all-cause dementia (*n* = 1490).

Endpoint	HR (95% CI) for Quintiles of Red Blood Cell EPA						HR (95% CI) per SD ^1^
	Q1 (<0.44%, Median = 0.37%) (*n* = 303)	Q2 (0.44% to <0.55%, Median = 0.49%) (*n* = 294)	Q3 (0.55% to <0.70%, Median = 0.61%) (*n* = 299)	Q4 (0.70% to 0.95%, Median = 0.78%) (*n* = 297)	Q5 (>0.95%, Median = 1.21%) (*n* = 297)	*p* for Trend	
AD							
N. of cases	39	31	42	27	29		168
Model 1	1.00	0.68 (0.42, 1.10)	0.97 (0.62, 1.51)	0.53 (0.32, 0.91)	0.74 (0.46, 1.20)	0.10	0.97 (0.81, 1.16)
Model 2	1.00	0.69 (0.42, 1.12)	0.99 (0.63, 1.55)	0.57 (0.33, 0.98)	0.76 (0.46, 1.25)	0.19	0.98 (0.81, 1.16)
Model 3	1.00	0.69 (0.42, 1.12)	0.98 (0.63, 1.55)	0.56 (0.32, 0.96)	0.74 (0.45, 1.23)	0.17	0.97 (0.81, 1.16)
Dementia							
N. of cases	32	24	31	22	22		131
Model 1	1.00	0.61 (0.35, 1.06)	0.84 (0.51, 1.38)	0.50 (0.28, 0.89)	0.66 (0.38, 1.14)	0.13	0.96 (0.78, 1.19)
Model 2	1.00	0.62 (0.35, 1.09)	0.90 (0.54, 1.50)	0.55 (0.30, 1.01)	0.69 (0.39, 1.24)	0.20	0.98 (0.79, 1.21)
Model 3	1.00	0.62 (0.35, 1.09)	0.90 (0.54, 1.50)	0.54 (0.29, 0.99)	0.68 (0.38, 1.21)	0.17	0.97 (0.78, 1.21)

^1^ 1 SD = 0.48% of total fatty acids. CI = confidence interval; Q = quintile; SD = standard deviation. See Table 2 for detailed information on statistical models.

**Table 5 nutrients-14-02408-t005:** Hazard ratios (HR) for omega-3 index on Alzheimer’s disease (AD) and all-cause dementia (*n* = 1490).

Endpoint	HR (95% CI) for Quintiles of Omega-3 Index						HR (95% CI) per SD ^1^
	Q1 (<4.3%, Median = 3.9%) (*n* = 300)	Q2 (4.3% to <5.1%, Median = 4.7%) (*n* = 300)	Q3 (5.1% to <5.9%, Median = 5.4%) (*n* = 298)	Q4 (5.9% to 7.1%, Median = 6.3%) (*n* = 293)	Q5 (>7.1%, Median = 8.0%)(*n* = 299)	*p* for Trend	
AD							
N. of cases	29	31	24	27	20		131
Model 1	1.00	0.81 (0.49, 1.36)	0.60 (0.34, 1.08)	0.74 (0.43, 1.29)	0.52 (0.28, 0.93)	0.04	0.83 (0.67, 1.02)
Model 2	1.00	0.90 (0.52, 1.54)	0.62 (0.34, 1.15)	0.81 (0.45, 1.45)	0.55 (0.29, 1.05)	0.08	0.85 (0.68, 1.05)
Model 3	1.00	0.90 (0.52, 1.54)	0.61 (0.33, 1.14)	0.81 (0.45, 1.45)	0.55 (0.29, 1.03)	0.07	0.84 (0.68, 1.04)
Dementia							
N. of cases	34	39	32	36	27		168
Model 1	1.00	0.90 (0.56, 1.44)	0.72 (0.43, 1.22)	0.88 (0.54, 1.44)	0.61 (0.36, 1.03)	0.09	0.87 (0.73, 1.04)
Model 2	1.00	0.93 (0.57, 1.50)	0.73 (0.42, 1.25)	0.93 (0.56, 1.54)	0.63 (0.36, 1.09)	0.14	0.88 (0.74, 1.05)
Model 3	1.00	0.93 (0.57, 1.51)	0.72 (0.42, 1.24)	0.92 (0.56, 1.53)	0.62 (0.36, 1.07)	0.12	0.88 (0.73, 1.05)

^1^ 1 SD = 1.70% of total fatty acids. CI = confidence interval; Q = quintile; SD = standard deviation. See Table 2 for detailed information on statistical models.

**Table 6 nutrients-14-02408-t006:** Theoretical effects of selected variables on event-free years of life (*n* = 1490).

Variable	Alzheimer’s Disease		All-Cause Dementia	
	Hazard Ratio	Scaled β	Hazard Ratio	Scaled β
Age, 1 year	1.16	1.00	1.15	1.00
*APOE-ε4* carriership, yes	3.12	−7.59	2.78	−7.30
RBC DHA, Q5 to Q1	2.01	−4.65	1.76	−4.03

HR = hazard ratio; RBC DHA = red blood cell docosahexaenoic acid; Q = quintile. Data based on age-adjusted models. Scaled β can be interpreted as the equivalent years of life free of AD and all-cause dementia associated with each variable and is estimated by taking the beta (ln HR) for each variable divided by the beta for age. Therefore, changing from RBC DHA from Q5 to Q1 would be associated with an estimated reduction of 4.65 years free of AD and 4.03 years free of dementia. Carrying an *APOE-ε4* allele would be associated with an estimated reduction of 7.59 and 7.30 years free of AD and all-cause dementia, respectively.

## Data Availability

Framingham Study data are available through BioLINCC, where qualified researchers can apply for authorization to access (biolincc.nhlbi.nih.gov/studies/framcohort/?q=Framingham, accessed on 8 June 2022).

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
