# Peer review of "Red Blood Cell DHA Is Inversely Associated with Risk of Incident Alzheimer’s Disease and All-Cause Dementia: Framingham Offspring Study"

_nutrients, 2022, doi:10.3390/nu14122408_

Round 1

Reviewer 1 Report

This is the nice report. Before publication, I have few comments to improve this documentation.

1) In the spline figures, what is the value of X axis?

Could you explain a bit more in the text?

2) If you have a example of fatty acid chromatography chart including DHA, please show in this paper.

3) Why the content of DHA is more effective in e4+ individuals? Could you explain the putative mechanism behind this very original finding, in the Discussion section?

Author Response

Reviewer #1. This is the nice report. Before publication, I have few comments to improve this documentation.

 1) In the spline figures, what is the value of X axis? Could you explain a bit more in the text?

The value of the X axis is the red blood cell proportion of DHA (as percentage of total identified fatty acids). Spline figures of the revised version of manuscript now include this information.

2) If you have an example of fatty acid chromatography chart including DHA, please show in this paper.

We included a chromatogram in the manuscript, labelling DHA. Please see new Figure 2.

3) Why the content of DHA is more effective in e4+ individuals? Could you explain the putative mechanism behind this very original finding, in the Discussion section?

We expanded the mechanisms behind our finding. In “Discussion”, please find: “We observed an interaction DHA × APOE-ε4 carriership on incident AD and all-cause dementia, with a trend towards a greater benefit of DHA in ε4 carriers than in non-carriers. A plausible explanation for this finding is that APOE-ε4 carriers might need more DHA to overcome their lower status of DHA (secondary to accelerated liver catabolism of DHA) coupled to impaired delivery of DHA to the brain [6]”.

Reviewer 2 Report

Thank you for the opportunity to review your manuscript entitled " Red blood cell DHA is inversely associated with risk of incident Alzheimer’s Disease and all-cause dementia: Framingham Offspring Study". This is a hot topic. Several association studies were found previous; however, the main strength of the present study is using objective measure of RBC DHA and relatively long f/u period of 7.2 years from well-design FHS. I think the results could have additional evidence in the field of research. Several comments are made to improve the manuscript.

1.      “If a participant had suspected cognitive impairment but did not meet criteria for a diagnosis of dementia, additional yearly neuropsychological assessments were performed between the quadrennial Offspring examinations”. Please clarify more regarding how to define “suspected cognitive impairment”. Did this based on any validated cognitive screening test?

2.      Why author divided DHA level into Q1 to Q5 but not Q1 to Q4? Please clarify this.

3.      Attrition rate is important to reduce selection bias. Author should mention this and compare those included in the study and those with lost f/u regarding the baseline demographic data.

4.      Provide information on the validity of the assumptions for the Cox proportional hazard model.

5.      The "survival plot" is interesting but should include confidence intervals and show the numbers at risk by group over time.

6.      Dementia has a long-term prodromal period, even decades. The mean follow-up period of the study was 7.2 years but wide range of 0-14 years. It is more likely for those diagnosed with dementia in the initial few years are undiagnosed dementia in the baseline (particularly the study included those aged over 65 years). To confirm the results, we suggest author to perform sensitivity analysis to exclude dementia cases within 2, 3, or 5 years.

7.      Please provide the rationale why author chose education, DM, cardiovascular disease in the model 2 and 3

8.      Because there are so many different confounders mediating the risk of dementia, I have some concerning regarding the chose of confounders. Specific comments are as follows:

(1)   Given the characteristic of offspring study, I am particularly interested in family history of dementia. For those with family history of dementia, they might be more likely to engage nutrition supplement, exercise, reduce possible unhealthy life habit. Therefore, it would be value to provide the information regarding family history of dementia and put in the adjusted model.

(2)   Also for confounders, few confounders provided in the table 1 such as DM, cardiovascular disease. Information regarding medications (e.g. anti-HTN, anti-DM, anti-lipid, aspirin etc.) between groups are recommended, and put in the adjusted model if needed.

9.      No association between outcome and omega-3 index (RBC DHA+EPA). Previous studies showed the ratio for EPA/DHA is important for the cognitive beneficial effect. I am curious if it is possible to perform analysis between the ratio of EPA/DHA and outcomes.

10.  The association between omega-3 (either fish intake, blood DHA levels, or RBC DHA) and reduced risk of dementia were reported previously. The clinical trials by giving DHA or EPA supplement to reduce the risk of dementia are failed. The author had already mentioned this in the 5th paragraph of discussion. Several updated RCTs are recommended to include in your discussion (PMID: 32690472, 34755655) (Note: I am not the author of these studies).

11.  Only once RBC DHA blood checking was done. This could be considered as one of limitations.

Author Response

Reviewer #2. Thank you for the opportunity to review your manuscript entitled " Red blood cell DHA is inversely associated with risk of incident Alzheimer’s Disease and all-cause dementia: Framingham Offspring Study". This is a hot topic. Several association studies were found previous; however, the main strength of the present study is using objective measure of RBC DHA and relatively long f/u period of 7.2 years from well-design FHS. I think the results could have additional evidence in the field of research. Several comments are made to improve the manuscript.

1. “If a participant had suspected cognitive impairment but did not meet criteria for a diagnosis of dementia, additional yearly neuropsychological assessments were performed between the quadrennial Offspring examinations”. Please clarify more regarding how to define “suspected cognitive impairment”. Was this based on any validated cognitive screening test?

We expanded this notion in the revised version of the paper. Please find: “Participants were flagged with suspected cognitive impairment using the Mini-Mental State Examination if (1) performance fell below education-based cutoff scores [23]; (2) a decline of 3 or more points was observed between consecutive examinations, or (3) a de-crease of 5 or more points was observed from the participant's highest past Mini-Mental State Examination score. Participants were also flagged following referrals or concern expressed by the participant, their family, or primary care physician”.

[23] Bachman, D.L.; Wolf, P.A.; Linn, R.; Knoefel, J.E.; Cobb, J.; Belanger, A.; D'Agostino, R.B.; White, L.R. Prevalence of dementia and probable senile dementia of the Alzheimer type in the Framingham Study. Neurology. 1992, 42, 115-119.

2. Why author divided DHA level into Q1 to Q5 but not Q1 to Q4? Please clarify this.

Assessment of hazard ratios in a per quintile basis has been commonly used in papers dealing with red blood cell fatty acids in this cohort. For example,  

- McBurney MI, Tintle NL, Vasan RS, Sala-Vila A, Harris WS. Using an erythrocyte fatty acid fingerprint to predict risk of all-cause mortality: the Framingham Offspring Cohort. Am J Clin Nutr. 2021 Oct 4;114(4):1447-1454. doi: 10.1093/ajcn/nqab195. PMID: 34134132; PMCID: PMC8488873.

- Block RC, Shearer GC, Holub A, Tu XM, Mousa S, Brenna JT, Harris WS, Tintle N. Aspirin and omega-3 fatty acid status interact in the prevention of cardiovascular diseases in Framingham Heart Study. Prostaglandins Leukot Essent Fatty Acids. 2021 Jun;169:102283. doi: 10.1016/j.plefa.2021.102283. Epub 2021 Apr 24. PMID: 33964664; PMCID: PMC8159885.

- Harris WS, Tintle NL, Ramachandran VS. Erythrocyte n-6 Fatty Acids and Risk for Cardiovascular Outcomes and Total Mortality in the Framingham Heart Study. Nutrients. 2018 Dec 19;10(12):2012. doi: 10.3390/nu10122012. PMID: 30572606; PMCID: PMC6316092.

- Harris WS, Tintle NL, Etherton MR, Vasan RS. Erythrocyte long-chain omega-3 fatty acid levels are inversely associated with mortality and with incident cardiovascular disease: The Framingham Heart Study. J Clin Lipidol. 2018 May-Jun;12(3):718-727.e6. doi: 10.1016/j.jacl.2018.02.010. Epub 2018 Mar 2. Erratum in: J Clin Lipidol. 2020 Sep - Oct;14(5):740. PMID: 29559306; PMCID: PMC6034629.

3. Attrition rate is important to reduce selection bias. Author should mention this and compare those included in the study and those with lost f/u regarding the baseline demographic data.

As has been noted in numerous publications (e.g., Tsao CW, Vasan RS. Cohort Profile: The Framingham Heart Study (FHS): overview of milestones in cardiovascular epidemiology. Int J Epidemiol. 2015;44(6):1800-1813; doi:10.1093/ije/dyv337), attrition in the Framingham Heart study is extremely low (<1% since study inception) and, thus, is not substantial enough in this analysis to warrant comparative analyses.

4. Provide information on the validity of the assumptions for the Cox proportional hazard model.

We have added a statement to the methods that Schoenfeld residuals were used to confirm the proportional hazards assumption.

5. The "survival plot" is interesting but should include confidence intervals and show the numbers at risk by group over time.

We have updated Figures 2a and 2b to add risk tables and confidence intervals.

6. Dementia has a long-term prodromal period, even decades. The mean follow-up period of the study was 7.2 years but wide range of 0-14 years. It is more likely for those diagnosed with dementia in the initial few years are undiagnosed dementia in the baseline (particularly the study included those aged over 65 years). To confirm the results, we suggest author to perform sensitivity analysis to exclude dementia cases within 2, 3, or 5 years.

Thank you for the suggestion. We have run these analyses (2, 3 and 5 years) and they show the same general pattern. We have included the 5-year results in the manuscript (please see new Table 3).

7. Please provide the rationale why author chose education, DM, cardiovascular disease in the model 2 and 3.

- Model 2: Besides a model including non-modifiable risk factors for AD/dementia (Model 1), we decided to construct an additional one including variables related to lifestyle / modifiable risk factors for AD. We included “education” because low education has long been identified as a risk factor for incident AD. For instance, in a systematic review by Sharp and Gatz (2011, PMID: 21750453), of the 14 studies that analyzed the relationship between low education and AD, 10 studies found a significant effect such that low education was associated with a significant increased risk for AD. We also included “prevalent DM at baseline” as a confounder because of shared molecular and cellular features among DM and AD (actually AD has even been called "type 3 diabetes").

- Model 3: This model was created to include the strong and likely causal association between cardiovascular disease (CVD – including sub-clinical CVD) and incident AD (actually the diseases often co-exist). The reviewer is kindly directed to Stampfer MJ (2006, PMID: 16918818).

8. Because there are so many different confounders mediating the risk of dementia, I have some concerning regarding the choice of confounders. Specific comments are as follows:

8.1-Given the characteristic of offspring study, I am particularly interested in family history of dementia. For those with family history of dementia, they might be more likely to engage nutrition supplement, exercise, reduce possible unhealthy life habit. Therefore, it would be value to provide the information regarding family history of dementia and put in the adjusted model.

In our opinion, given our focus on genetic risk factors (e.g., APOE4 status) additional consideration of family history would likely constitute overfitting of the current models. Furthemore, a more in-depth look at additional genetic risk factors is outside of the scope of this work. Other health behaviors are captured at the individual level via existing covariates.

8.2-Also for confounders, few confounders provided in the table 1 such as DM, cardiovascular disease. Information regarding medications (e.g. anti-HTN, anti-DM, anti-lipid, aspirin etc.) between groups are recommended, and put in the adjusted model if needed.

This is a good point. When designing the statistical analysis, we discussed the benefits of including these variables in model 2. Given the sample size and the number of incident cases, coupled to the presumed need to stratify for APOE-ε4 carriership, we decided to limit the variables included as confounders to avoid overfitting of the model.

9. No association between outcome and omega-3 index (RBC DHA+EPA). Previous studies showed the ratio for EPA/DHA is important for the cognitive beneficial effect. I am curious if it is possible to perform analysis between the ratio of EPA/DHA and outcomes.

The study was conceived to explore the presumed positive associations for DHA, the main omega-3 in brain. We acknowledge that the issue of whether EPA and DHA act in a synergistic way is a topic of current interest, being explored in several randomized controlled trials in patients with AD (Song C et al., 2016, PMID: 26763196). However, we believe that presumed associations between the ratio EPA/DHA and incident AD might be misleading, since this ratio can be modified by either increasing DHA intake (something that might have a truly benefit) or decreasing EPA intake (the benefit of which is less clear).

10. The association between omega-3 (either fish intake, blood DHA levels, or RBC DHA) and reduced risk of dementia were reported previously. The clinical trials by giving DHA or EPA supplement to reduce the risk of dementia are failed. The author had already mentioned this in the 5th paragraph of discussion. Several updated RCTs are recommended to include in your discussion (PMID: 32690472, 34755655) (Note: I am not the author of these studies).

The revised version of the manuscript refers to both trials, as well as others. Please find:

  1. Dangour, A.D.; Allen, E.; Elbourne, D.; Fasey, N.; Fletcher, A.E.; Hardy, P.; Holder, G.E.; Knight, R.; Letley, L.; Richards, M.; Uauy, R. Effect of 2-y n-3 long-chain polyunsaturated fatty acid supplementation on cognitive function in older people: a randomized, double-blind, controlled trial. Am J Clin Nutr. 2010, 91, 1725-1732.
  2. Chew, E.Y.; Clemons, T.E.; Agrón, E.; Launer, L.J.; Grodstein, F.; Bernstein, P.S.; Age-Related Eye Disease Study 2 (AREDS2) Research Group. Effect of Omega-3 Fatty Acids, Lutein/Zeaxanthin, or Other Nutrient Supplementation on Cognitive Func-tion: The AREDS2 Randomized Clinical Trial. JAMA. 2015, 314, 791-801.
  3. Kang, J.H.; Vyas, C.M.; Okereke, O.I.; Ogata, S..; Albert, M.; Lee, I.M.; D'Agostino, D.; Buring, J.E.; Cook, N.R.; Grodstein, F.; Manson, J.E. Marine n-3 fatty acids and cognitive change among older adults in the VITAL randomized trial. Alzheimers De-ment (N Y). 2022, 8,e12288.
  4. ...
  5. Lin, P.Y.; Cheng, C.; Satyanarayanan, S.K.; Chiu, L.T.; Chien, Y.C.; Chuu, C.P.; Lan, T.H.; Su, K.P. Omega-3 fatty acids and blood-based biomarkers in Alzheimer's disease and mild cognitive impairment: A randomized placebo-controlled trial. Brain Behav Immun. 2022, 99, 289-298.
  6. ...
  7. ...
  8. Arellanes, I.C.; Choe, N.; Solomon, V.; He, X.; Kavin, B.; Martinez, AE.; Kono, N.; Buennagel, D.P.; Hazra, N.; Kim, G.; D'Ora-zio, L.M.; McCleary, C.; Sagare, A.; Zlokovic, B.V.; Hodis, H.N.; Mack, W.J.; Chui, H.C.; Harrington, M.G.; Braskie, M.N.; Schneider, L.S.; Yassine, H.N. Brain delivery of supplemental docosahexaenoic acid (DHA): A randomized place-bo-controlled clinical trial. EBioMedicine. 2020, 59:102883.

11. Only once RBC DHA blood checking was done. This could be considered as one of limitations.

This limitation has been included in the “Discussion”. Please find: “Fourth, there is no information on whether a single measurements of RBC DHA is appropriate to estimate the risk of AD over long-term follow up when compared to repeated measurements”.

Round 2

Reviewer 2 Report

No further comments.

Thank you.

Author Response

Thanks for your review.